# The use of digital texture image analysis in determining the masticatory efficiency outcome

**Aleksandra Milić Lemić** [1][*], **Katarina Rajković**[2], **Katarina Radović**[1‡], **Rade Živković**[1‡], **Biljana Miličić**[3], **Mirjana Perić**[1‡]

**1** Clinic for Prosthetic Dentistry, Faculty of Dental Medicine, University of Belgrade, Belgrade, Serbia, **2** College of Applied Studies of Technics and Technology, Kruševac, Serbia, **3** Department for Informatics and Biostatistics, Faculty of Dental Medicine, University of Belgrade, Belgrade, Serbia

☯ These authors contributed equally to this work.
‡ These authors also contributed equally to this work.
* aleksandra.milic@stomf.bg.ac.rs

**Data Availability Statement:** All relevant data are within the manuscript and its Supporting information files.

## Abstract

The mixture level of gum samples consisting of two colours can be assessed visually, using the electronic colorimetric method, employing digital image processing techniques and specially designed software. The study investigates the possibility of an alternative method called "digital texture image analysis" (DTIA) to assess improvement of masticatory efficiency in denture wearers. The objectives were i) to evaluate whether DTIA discriminates changes in the colour mixing ability within a group over time; ii) to determine whether DTIA can be used to detect improvement in chewing ability; iii) to select the most appropriate DTIA feature that sufficiently describes masticatory efficiency in CDs wearers. The study was designed as an intra-individual evaluation of masticatory efficiency, which was assessed in participants with new dentures in three follow-up times. A set of four texture features was used in the current study. Uniformity, Contrast, Homogeneity and Entropy of the obtained chewing-gum samples were correlated to the degree of gum comminution. A statistically significant difference in masticatory efficiency was observed based on the values of the analysed DTIA variables of gum samples—Uniformity, Contrast, Homogeneity, and Entropy—have changed in the participants during the observation period. The improvement of the masticatory function in relation to the mixing ability of two-coloured chewing gum could be traced by monitoring changes in the values of DTIA variables. The most increasement of masticatory efficiency was observed by monitoring DTIA parameters such as contrast, and homogeneity.

## Introduction

It is indisputable that tooth loss and edentulism have an impact on masticatory function and a consequent influence on the nutritional status [1–3]. Although the prevalence of edentulism has fallen markedly in industrialised countries [4, 5], tooth loss is among the main conditions

**Funding:** The author(s) received no specific funding for this work.

**Competing interests:** The authors have declared that no competing interests exist.

of interest when considering oral diseases among the elderly [6], who are already affected by other systemic diseases. Deterioration of masticatory function affects the efficiency of food absorption and nutritional status [7], limiting their food selection and dietary intake of fibres, magnesium or calcium [8, 9], with potential effect on systemic health.

Edentulous people are likely to be treated with complete dentures (CDs) in order to rehabilitate their masticatory function to some extent. The ability to successfully wear a pair of CDs is, to an extent, a learned, skilled performance. Thus, it is expected that aged subjects who are first-time CDs wearers may need time to cope with a new pair of dentures.

After insertion of complete dentures, the chewing ability improves as patients get accustomed to their new dentures, and the improved fit optimises retention and stability within the limits allowed by the anatomical and tissue conditions [10]. Chewing ability expressed as masticatory efficiency is a dynamic chewing function and it is objectively evaluated by observing, measuring or quantifying food changes during mastication [11]. Various methods characterized as objective have been used in masticatory efficiency assessment so far, where researchers evaluated changes in chewed natural foods [12], or synthetic food [13], as in sieving methods. The sieving method, which measures the comminution of test materials, was considered the gold standard for assessing the masticatory efficiency, especially the multiple sieve method, which yields better results than the single one [14]. A substitution of comminution tests is the ability to mix and knead a food bolus, usually simulated by two-coloured chewing gum [15, 16] or paraffin wax [17, 18], which was also widely used in masticatory efficiency evaluation [19]. The degree of mixing of two colours was determined by optical methods [17, 19], by visual inspection [20] or by both [21]. Later, software was introduced for image processing of two-colour chewing gum mixing [22]. Colorimetric detectability of chewing-gum colour changes has proven to be as reliable method in clinical and academic masticatory efficiency assessment. These methods allow for the evaluation of gum colour changes but lack deeper insight into the changes of gum sample texture appearance. We hypothesized that using the "digital texture image analysis" (DTIA) might provide a better insight into the improvement of masticatory efficiency in denture wearers. DTIA is based on the grey level co-occurrence matrix (GLCM) method, where the texture of an image corresponds to the spatial organisation of pixels in the image, and the co-occurrence matrix describes the occurrence of grey level between two pixels separated in the image by a given distance [23]. There are up to 14 textural features of DTIA which may be selected to represent the textural characteristics of the image under investigation [23].

The DTIA method was used to evaluate the structure of osteoporotic bone based on a series of images [24], to follow the periapical bone healing [25] or differentiate soft tissue lesions [26]. Also, texture image analysis was used to characterise the impact of chewing on pasta particle size reduction [27] or food bolus characterisation [28]. Although used sporadically in dental research, there are no publications about the application of DTIA in assessment of the digital images of chewed two-colour gums during the mixing ability test in masticatory efficiency evaluation.

Therefore, we investigated the applicability of DTIA in evaluating and measuring the bolus-kneading capacity of two-coloured chewing gum samples in complete denture wearers. The objectives were i) to evaluate whether DTIA discriminates changes in the colour mixing ability within the group in follow-up time; ii) to determine whether DTIA can be used to detect improvement in chewing ability iii) to select the most appropriate DTIA feature that sufficiently describes masticatory efficiency in CDs wearers.

## Materials and method

The study was designed as a prospective intra-individual cohort investigation of masticatory efficiency in edentulous subjects with new CDs lasting over 6 months. Following the requirements of the Declaration of Helsinki on ethical principles for medical research involving human subjects and according to the STROBE guidelines, prior to the very beginning of the investigation the study protocol was approved by the Ethics Committee of the School of Dental Medicine, University in Belgrade (No 35/14). All participants were given a detailed explanation of the purpose and process of the study and gave their written informed consent. The investigation was conducted from October 2017 to July 2018 (including recruitment, clinical procedures, follow-up after three and six months, and data collection).

### Inclusion and exclusion criteria

Participants were recruited from a pool of patients who visited the Clinic of Prosthodontics, asking for prosthodontics rehabilitation.

The basic inclusion criterion was that the subjects in the study be of both sexes, over 65 years and edentulous. The specific inclusion criterion was related to the quality of the prosthesis supporting tissues. After clinical suitability examination, the subjects with the Kapur index higher than 14 for the mentioned parameters were included in the study [29, 30]. After the complete dentures were made, their clinical quality was analysed. In order for a subject to be included in the study, it was necessary that at the clinical examination a pair of complete dentures meet the criterion for good clinical quality of dentures and have the Kapur index higher than 6 [29, 31].

The exclusion criteria were the presence of signs of TMJ dysfunctions or neuromuscular disorders. Participants who were unable to use the dentures continuously or who were unwilling to attend the follow-up appointments were excluded from the study, as well.

Based on the defined inclusion / exclusion criteria, after a clinical suitability examination and fabrication of upper and lower CDs, the study group consisted of 20 subjects. However, for private reasons, two subjects withdrew and the definitive number of subjects in the study group was 18.

After the insertion of a new pair of CDs, verbal and written instructions about CDs insertion, removal, cleaning, and care were given to all participants.

### Test sample and chewing protocol

Masticatory efficiency was assessed according to the described study protocol, by measuring the bolus-kneading capacity through the ability to mix two-coloured chewing gum [32]. The study group chewing gum samples were obtained at three different time periods, and three subgroups of samples were formed: at the baseline (T0), after 3 months (T1) and after 6 months (T2).

The samples for testing were prepared from the Five Tape sugarless gum (The Wrigley Company Ltd, Plymouth, Devon, PL6 7PR, England) in the following flavours: peppermint (green colour) and strawberry (pink colour). Dimensions of the chewing gum samples were adopted from the literature [15, 21].

Thirty mm long strips were cut out of the green chewing gum and pink chewing gum. The strips of different colours were then joined lengthwise to form a sample with two different colour layers, of the overall dimensions of 30mmx18mmx3mm.

## Clinical protocol and image assessment

Upper and lower CDs were inserted and a sample of gum was placed onto the patient's tongue with the pink side oriented towards the palate. Each subject was instructed to sit upright and to chew the sample of gum on the preferred chewing side for 20 cycles [15]. The chewing cycles were evaluated by the operator, who counted the rhythmic open-close movement of the lower jaw in the anterior plane as one chewing stroke [33, 34].

Every participant chewed two chewing gum samples, according to the described procedure; therefore, there was a total of two samples per subject for analysis. Between two consecutive chewing cycles an interval of 1 min was imposed in order to minimize the effect of fatigue. Immediately after the final chewing cycle, the samples were placed directly into a transparent plastic bag labelled with an identification code. Each specimen was compressed to a 1mm-thick piece using a custom-made polyvinyl chloride plate with a milled depression according to the described protocol [15]. Both sides of the piece were photographed with a Nikon D3200 digital camera using 18–55 mm lens and Macro filter 8x, (Nikon Corp. Japan) and saved in JPEG format as described in the literature [32].

**Digital texture image analysis (DTIA).** The DTIA procedure was adopted from the literature [23–26] and applied to the gum samples in this study. Each chewed specimen was analysed from both sides, so that the images of both sides of the gum were converted from red, green, and blue (RGB) images (Fig 1A) to grayscale (GS) images (Fig 1B) by applying Image J software v 1.34s (National Institutes Health, Bethesda, MD, USA).

Later, the Grey Level Co-Occurrence Matrix (GLCM) algorithms were applied to obtain texture features of the gum images under investigation [23]. GLCM describes the second-order statistics in the images, enables a calculation of the textural characteristics, compares two neighbouring pixels at a time and compiles the frequency at which different grey-levels can be found within a restricted area [35, 36]. Based on the described operations, the software automatically generates values for the selected parameters. In this study, homogeneity, contrast, uniformity and entropy were selected as independent variables that describe the level of masticatory performance.

Uniformity represents the texture uniformity of the image and it is the opposite feature of entropy. It measures the number of repeated pairs, which is expected to be high if the occurrence of repeated pixel pairs is high [37]. Contrast is a measure of local variations of grey level values of pixels in the images [36]. Homogeneity measures the local homogeneity of a pixel pair. Homogeneity is expected to be large if the grey levels of each pixel are similar [37]. Entropy measures the disorder, or randomness of images, and can be used to characterize the image texture. It is an indicator of the complexity within an image, so, the more complex the images, the higher the entropy values [36, 38, 39]. A small difference in the levels of the grey scale of the pixels in the image corresponds to lower values of contrast and entropy, which means better colour mixing and consequently better masticatory performance. Also, small differences in the levels of the grey scale of the pixels in the image are in relation to higher values of uniformity and homogeneity, which means better colour mixing and consequently efficient masticatory performance. The digital texture features used were also described by Tournier et al. [23].

**Colorimetric analysis.** Colorimetry (analysis of variance of the hue) was performed using the freeware ViewGum© software (dHAL Software, Greece, www.dhal.com), and a step-by-step procedure for delimitation of the gum images, and VOH calculation was adopted from the literature [22]. The obtained images of specimens were imported into the custom ViewGum software, after which the software automatically executed the segmentation process. As seen in Fig 1A, after segmentation, the images were transformed from RGB values (red, green,

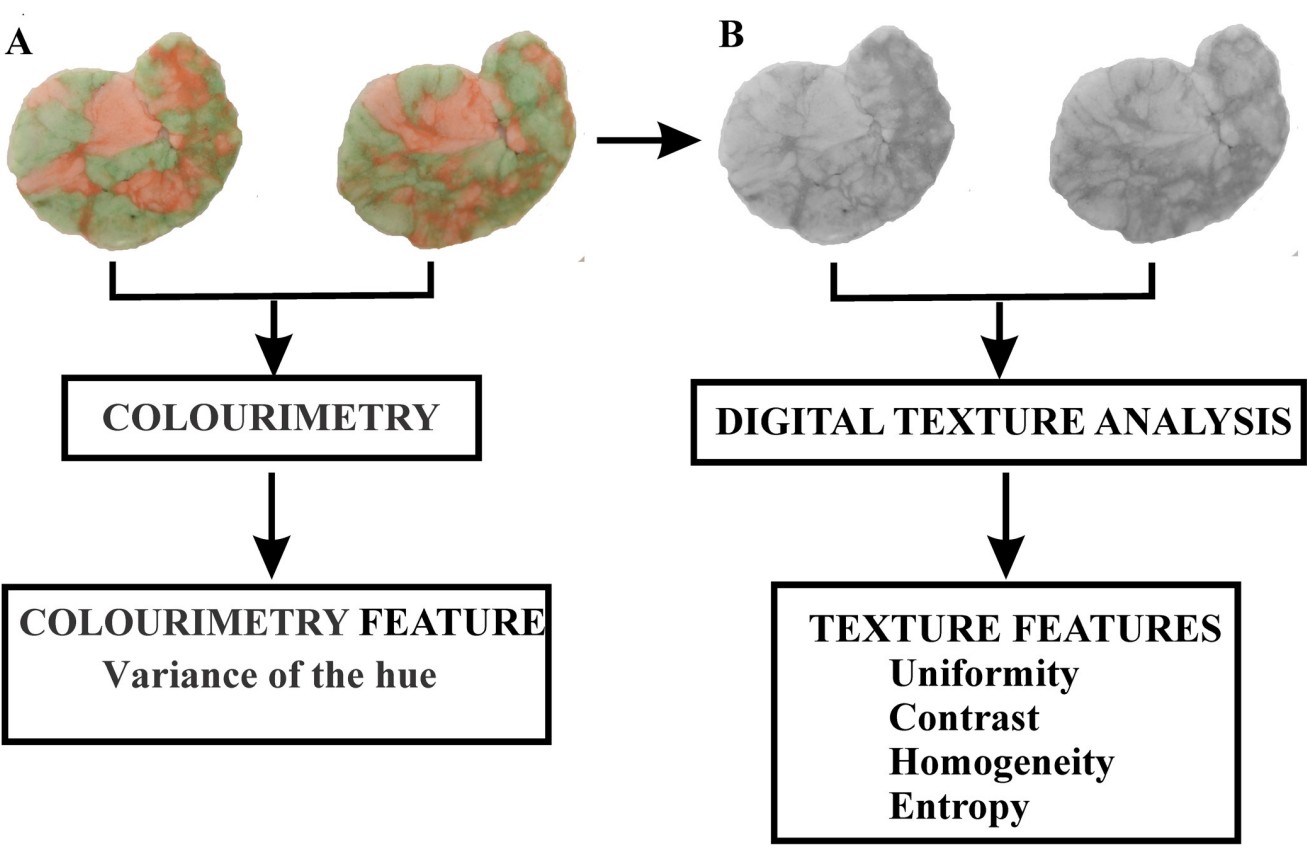

**Fig 1. The schematic presentation of the image processing procedure.** The red, green, and blue images (RGB) and the grayscale (GS) images.

blue) into the HSI colour space (hue, saturation, intensity) using an image processing application to analyse separately the hue, intensity and saturation [32]. A more representative measure of mixing two colours is achieved with only the hue component [22]. When the two colours of the chewing gum are not well mixed, then the neighbouring pixels belong to groups of different hues, with higher values. Therefore, the variance of hue (VOH) was set as the measure of level of colour mixture [21], which represents fusing the two colours into a single colour, where the lower VOH value corresponds to better colour mixing and consequently better masticatory performance.

## Statistical analysis

Statistical analyses were performed using SPSS software (IBM SPSS Statistics 2019 Version 26, USA). All obtained values were numerical. Descriptive data were expressed as mean, standard deviation, median and the interquartile range. Numeric data were tested for normal distribution using the Koglomorov-Smirnov test. All obtained data were non-parametric; therefore, the Fridman test and Wilcoxon test were used to analyse differences between the data at different follow-up times and baseline. This analysis tested the difference in the values of the obtained parameters measured on the upper and lower side of the chewing gum sample immediately after receiving complete dentures, after three months, and after six months. The analysis of the parameters measured on the upper side of the gum evaluated the adequacy of dental restorations in the upper and lower jaw. The Mann Whitney test was used for intra-individual

comparison, between the upper and lower side of the chewing gum sample. Differences were considered significant when p-value was <0.05.

## Results

### Participants

After all, the study included 18 participants with the mean age of 68 ± 4,6 (ranging from 65 to 80 years), and all of them completed the follow-up visits. All obtained chewing samples (N = 108) were suitable for further evaluation. The representative samples of chewing gum mixing ability images for $T_0$, $T_1$, and $T_2$ are shown in Fig 2A–2C.

Each patient chewed two samples of chewing gum, so there were 36 chewed gum samples per subgroup with a total of 108 chewed samples. A flow diagram of participants and total gum samples is presented in Fig 3.

The main variables of investigation are presented in Table 1, expressed as mean, standard deviation, median and the interquartile range. Based on the intra-group comparison between the chewed samples of both cycles, significance was observed in the Uniformity group at the baseline and after three months, for Entropy at the baseline, and after three months for VOH. Furthermore, from the values presented in Table 1, one may say that the values of the analysed variables of gum samples—Uniformity, Contrast, Homogeneity, and Entropy—have changed in the participants during the observation period, from the moment of CDs delivery (T0), after three months (T1), and after six months (T2). Also, a statistically significant change of VOH in the tested gum samples was noticed for all three follow-up times, which can also be seen in Table 1.

Similarity between the upper and lower side of the gum samples is observed in the Contrast and Homogeneity values for all three follow-up times, as seen in Table 1, while differences between the chewing gum samples are found for Entropy in $T_1$. The values for Uniformity measurements of one subject were significantly different at the beginning and three months after, and for VOH after three months.

Further analysis of the results obtained included the Wilcoxon test, a comparison between different measurement times, as shown in Table 2. Furthermore, statistically significant changes in the values are seen, especially when comparing the parameters Uniformity, Contrast and Homogeneity at the baseline (T0), with regards to the values after three months (T1) and after six months (T2). More graphical representations of the variable values are seen in Figs 1–10 in S1 Appendix. When observing the VOH values of the examined samples in all three time periods, statistically significant differences between the baseline (T0), after three months (T1) and after six months (T2) were noticed, as well as between the values of T1 and T2 (Figs 9 and 10 in S1 Appendix).

## Discussion

### Summary of the results

Implementing DTIA enabled a thorough masticatory efficiency analysis, with discrimination of masticatory efficiency progression within the edentulous subjects during adaptation time. The obtained results give insight into how fast the process of adaptation is going in edentulous patients who are first-time CDs wearers in relation to masticatory efficiency. The implemented method of DTIA is new and it has been supplemented with the already published and accepted colour mixture VOH analysis [15, 21, 22, 34]. Although they analyse different aspects of chewed gum samples, both analyses proved masticatory efficiency, discriminative to the analysed study group during the observed time, and DTIA was found to be consistent with the

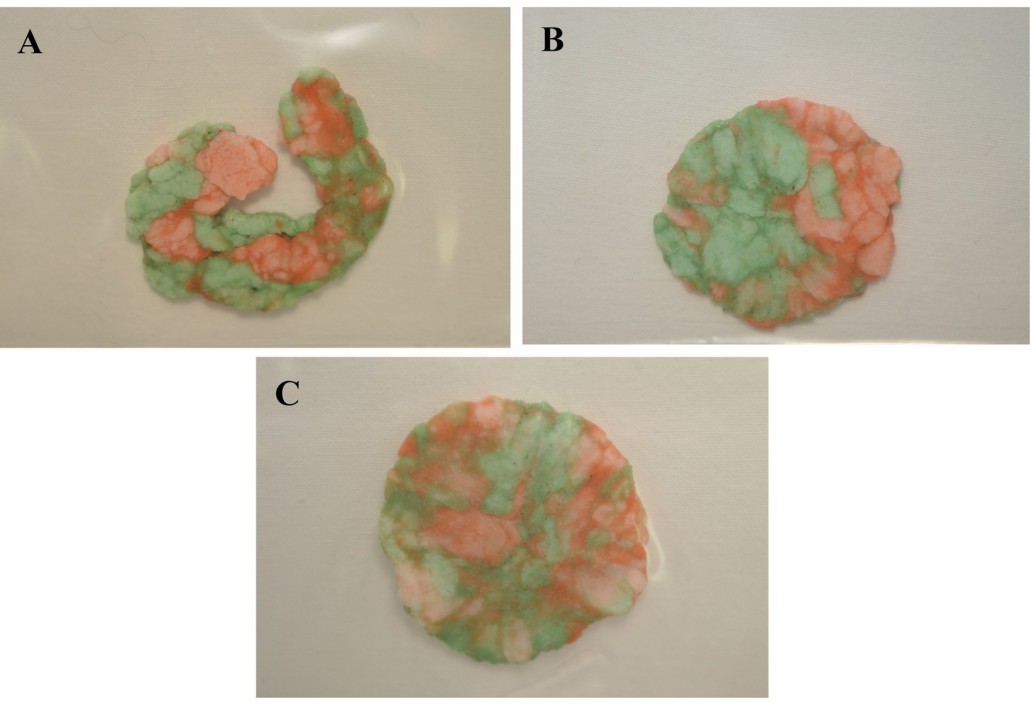

**Fig 2. Representative samples of chewing gum mixing ability images for T0, T1, and T2.**

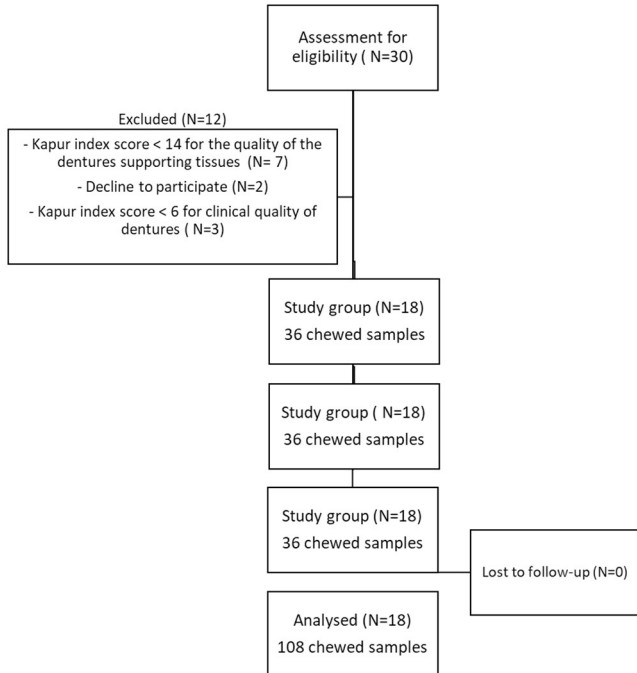

**Fig 3. Flow diagram of participants and total gum samples implemented in the study.**

**Table 1. Parameter values in the study group at the baseline, after three and after six months.**

| Variables | Baseline (T0) N = 18 | After three months (T1) N = 18 | After six month (T2) N = 18 | #Significance |
|---|---|---|---|---|
| Uniformity (upper side) | 0.0033 (0.000575) 0.0030 (0.000) | 0.0028 (0.0008) 0.0030 (0.001) | 0.0031 (0.00024) 0.0030 (0.0000) | [a]p = 0.010 |
| Uniformity (lower side) | 0.0034 (0.000575) 0.0033 (0.000) | 0.0027 (0.0003) 0.0028 (0.00022) | 0.0030 (0.0008) 0.0030 (0.001) | [a]p = 0.001 |
| [&] Significance | [b]p = 0.029 | [b]p = 0.001 | [b]p = 0.318 | |
| Contrast (upper side) | 2.870 (0.641) 2.645 (0.719) | 2.365 (0.414) 2.273 (0.262) | 2.098 (0.356) 2.043 (0.511) | [a]p = 0.001 |
| Contrast (lower side) | 2.676 (0.471) 2.773 (0.196) | 2.337 (0.116) 2.351 (0.108) | 2.168 (0.288) 2.133 (0.272) | [a]p = 0.001 |
| [&] Significance | [b]p = 0.974 | [b]p = 0.142 | [b]p = 0.508 | |
| Homogeneity (upper side) | 0.539 (0.024) 0,542 (0,030) | 0.560 (0.019) 0.566 (0.022) | 0.576 (0.025) 0.581 (0.038) | [a]p = 0.001 |
| Homogeneity (lower side) | 0.547 (0.010) 0.543 (0.043) | 0.564 (0.008) 0.562 (0.006) | 0.569 (0.020) 0.572 (0.018) | [a]p = 0.002 |
| [&] Significance | [b]p = 0.420 | [b]p = 0.936 | [b]p = 0.497 | |
| Entropy (upper side) | 6.24 (0.13) 6.26 (0.173) | 6.326 (0.138) 6.31 (0.124) | 6.208 (0.086) 6.205 (0.134) | [a]p = 0.001 |
| Entropy (lower side) | 6.12 (0.096) 6.07 (0.067) | 6.346 (0.067) 6.33 (0.089) | 6.222 (0.086) 6.215 (0. 047) | [a]p = 0.001 |
| [&] Significance | [b]p = 0.006 | [b]p = 0.323 | [b]p = 0.386 | |
| VOH (upper side) | 0.062 (0.062) 0.050 (0.095) | 0.129 (0.026) 0.127 (0.037) | 0.068 (0.024) 0.066 (0.034) | [a]p = 0.001 |
| VOH (lower side) | 0.076 (0.059) 0.068 (0.090) | 0.104 (0.026) 0.116 (0.052) | 0.058 (0.006) 0.062 (0.011) | [a]p = 0.005 |
| [&]Significance | [b]p = 0.235 | [b]p = 0.020 | [b]p = 0.299 | |

Mean (SD) and Median (IQR) values of the parameter at the baseline, after three and after six months.

[#]comparison between follow-up periods;

[&:]intergroup comparison;

[a] Fridman test;

[b]Mann Whitney test.

**Table 2. Values for intergroup comparison.**

| Variables | Follow-up time | T0 | T1 |
|---|---|---|---|
| Uniformity (upper) | T1 | p = 0.052 | / |
| | T2 | p = 0.248 | p = 0.102 |
| Uniformity (lower) | T1 | p = 0.002 | / |
| | T2 | p = 0.296 | p = 0.032 |
| Contrast (upper) | T1 | p = 0.001 | / |
| | T2 | p = 0.001 | p = 0.001 |
| Contrast (lower) | T1 | p = 0.016 | / |
| | T2 | p = 0.000 | p = 0.030 |
| Homogeneity (upper) | T1 | p = 0.002 | / |
| | T2 | p = 0.010 | p = 0.004 |
| Homogeneity (lower) | T1 | p = 0.001 | / |
| | T2 | p = 0.001 | p = 0.443 |
| Entropy (upper) | T1 | p = 0.048 | / |
| | T2 | p = 0.459 | p = 0.002 |
| Entropy (lower) | T1 | p = 0.001 | / |
| | T2 | p = 0.001 | p = 0.001 |
| VOH (upper) | T1 | p = 0.001 | / |
| | T2 | p = 0.679 | p = 0.001 |
| VOH (lower) | T1 | p = 0.048 | / |
| | T2 | p = 0.500 | p = 0.001 |

[*]statistical significance; Wilcoxon test.

colorimetric method, which was considered the gold standard. Also, DTIA possesses good reliability for two texture features, since there was similarity between the samples from the same subject for Contrast and Homogeneity. Therefore, Contrast and Homogeneity might be identified as a textural feature marker that could quantify chewing ability.

## Strength and weakness of the study

Although some authors consider the sieving test to be the most reliable method of mastication analysis [32, 40], the chewing gum test was reported to be as reliable for quantifying masticatory performance [30], especially in CD wearers [16, 19, 32] with compromised mastication, as was the case in the conducted study. This type of test has many advantages: reduced possibility of swallowing food parts and, consequently, loss of analytical material, as well as the fact that the elasticity of the material allows for the use of maximum available mastication capacity. Also, the mixing-ability tests for masticatory efficiency are less dependent on the flow rate of saliva, which is of particular importance in older people who often suffer from dry mouth syndrome [40]. The duration of 20-cycle chewing was implemented in the study according to the recommendations in the literature, as this seems to sufficiently discriminate between the 'physiological' chewing efficiency of a young, healthy and fully dentate person and an edentulous elderly individual with a substantial masticatory handicap [15]. The analysis of the inter and intra-individual reproducibility suggested that, for the application of the test, a fixed number of 20 chewing cycles should be applied. Also, it is reported that, when groups of participants are to be correlated, it is better to analyse 20 strokes, since it is not known after how many strokes the comparison curves keep diverging [41]. This number of chewing cycles was found useful, whereas high repeatability allows for follow-up evaluations, for example during a dental treatment or neuromuscular rehabilitation programmes [21].

It is reported that a significant increase in masticatory efficiency can be obtained about 6 weeks after the insertion of new complete dentures, while some authors stated that biting abilities improved 2 months after the insertion of new prostheses [42, 43]. Nevertheless, we performed the first follow-up after three months. The idea was to give more time to the new wearers to get used to dentures and to perform a more realistic mixing ability test. The masticatory function test was performed 6 months after the fabrication of good-quality complete dentures, as it was reported that during that period new muscle memory patterns for mastication would have been established [44].

Edentulous patients with the adequate size of the residual alveolar ridge have been included in this study, in order to avoid the impact of residual ridge resorption on masticatory function. However, it was recently indicated that mandibular bone atrophy does not interfere directly with the masticatory function of edentulous patients [45]. Confounding factors have been minimized since the impact of ill-fitting dentures, improper denture bearing tissues and other factors were eliminated, allowing the cause (edentulism) and effect (masticatory efficiency) to be distinguished.

The limitations of this study may be the small sample size, as well as the shortcomings of chewing tests. However, the evaluation of the literature dealing with similar topics revealed similar sample sizes [41–43]. One drawback of the present study is the missing control group in the form of dentate subject. However, findings for the dentate are well documented elsewhere but data about edentulous subjects are lacking. The main shortcoming of the study may be the use of commercial chewing gum in the analysis of masticatory efficiency and not of gum specially designed for this purpose. However, commercial chewing gum was used elsewhere in the literature as specimen for the chewing gum mixing ability test [21, 46]. Also, when testing CDs wearers using this method, one must take into account the possibility of the

onset of irritation and pain caused by wearing dentures during the adaptation period, which may affect the masticatory efficiency of these patients. Hence, transitory problems such as discomfort, sore spots, and injured mucosa were taken care of at the very beginning, providing minimum impact on the conducted investigation.

## Interpretation of the results in the context of applied methodology

The analysis of the literature showed that, for the analysis of the mixture level between the two colours of the gum, samples might be assessed both visually and by electronic colorimetric assessment [32], or by using digital image processing techniques [15] and specially designed software [22]. The evaluation of the obtained results in the mentioned procedures is more or less based on the measurement of fusing of the original colours into a new colour, with different intermediate intensities.

As opposed to the mentioned studies, we implemented DTIA, which gave insight not only into the colour change, but into the texture of the digital gum image. The texture refers to a visual feature which is homogenous, but the homogeneity does not come from one colour. The texture features used in the study are variables such as uniformity, contrast, homogeneity and entropy of the obtained chewing gum sample images, which are in direct relation to the comminution of the gum, and their values for each participant are presented in Fig 4. Uniformity refers to the existence of irregularities within the surface of the specimen colour image, and there will be fewer of them when the sample is better blended / chewed. Contrast measures differences in the colour of the specimen, which means that it is reduced in well chewed samples, where the colours are well mixed. Homogeneity, in the context of measuring chewing efficiency, shows that a mixture of two colours was created, meaning the sample has been mixed and the colours merged into one. Entropy is a measure of the image system regulation of the specimen. The smaller it is, the sample is more arranged, or rather—better chewed.

The Uniformity values are the lowest at the first follow-up, as seen in Fig 4A, showing the chewiness of the sample after the third month. This suggested that the participants at the baseline faced with the chewing gum samples between CDs masticated vigorously, thus confirming the initial uniformity inside the sample. Later, after a certain period of use ($T_2$), when analysed again, their mastication was more coordinated, not requiring vigorous mastication. These findings should be confirmed in future research with parallel evaluation of masticatory force values.

The evaluation of the obtained results shows that the easiest method of observing the masticatory efficiency in edentulous subjects is by analysing Contrast and Homogeneity. Contrast, as a measure of chewed samples, shows a decrease during the observation period, as seen in Fig 4B. This means that the contrast in the tested samples is reduced, due to higher masticatory efficiency and more complete chewing. The values for Contrast were highest at the baseline, confirming that mixing ability is scarce, with distinguished colours in the samples and great contrast between the colours. Contrast among all tested texture features showed compliance between the test-retest samples in all three follow-ups. The same is found for the Homogeneity feature, which possesses good reliability, since there is also similarity between the test-retest samples of the same subject. Tournier et al. [23] arrived at the same conclusion, which showed that contrast could be used successfully to investigate food bolus formation during chewing different breads.

The Homogeneity parameter values are increasing from $T_0$ toward $T_2$ with statistical significance (Table 1), showing a gradual improvement of masticatory efficiency. The graph in Fig 4C shows an increase of masticatory efficiency in terms of homogeneity after three months

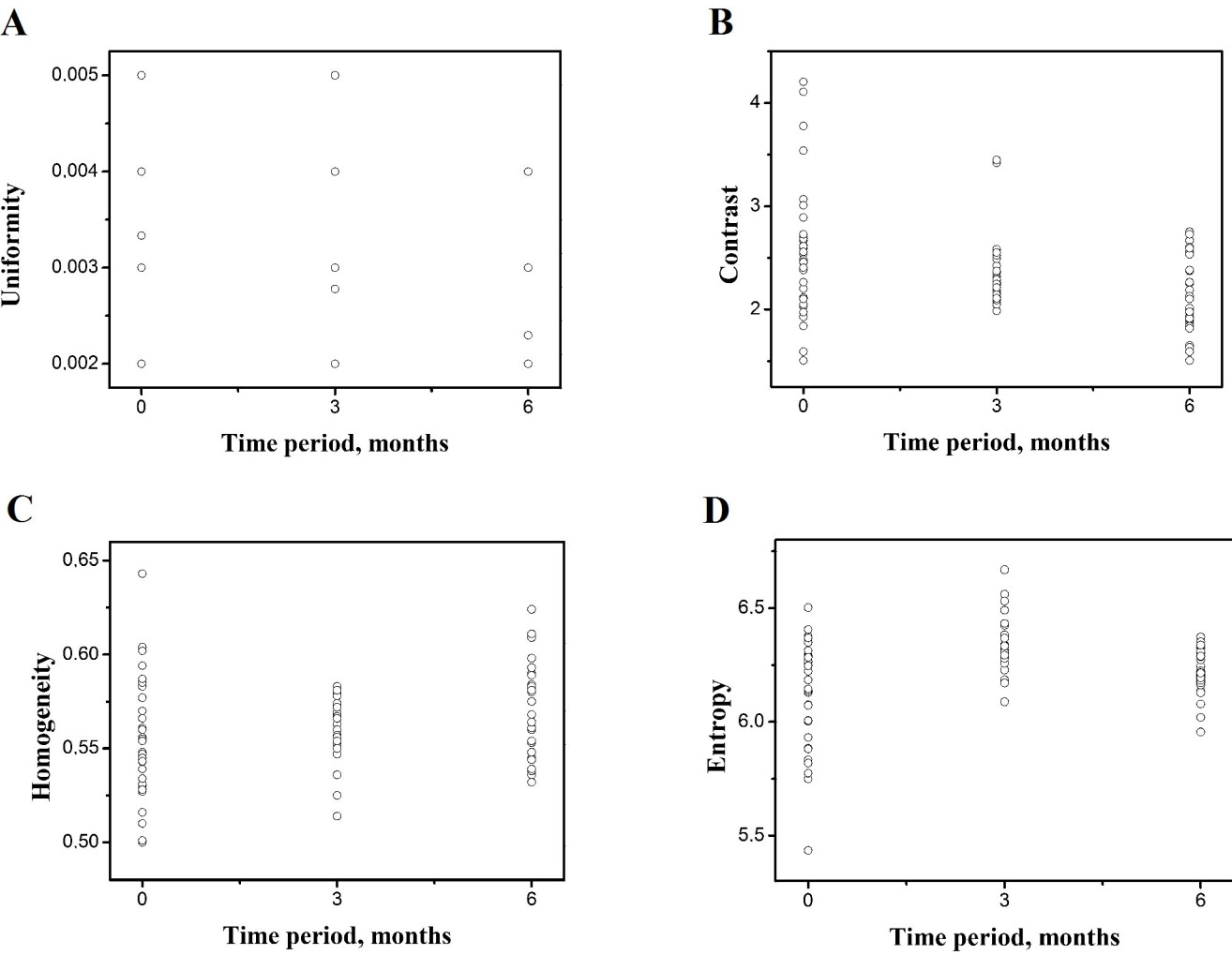

**Fig 4. DTIA variable values for each patient during the observation period (at the baseline, after three months, after six months).** Uniformity (A); Contrast (B) Homogeneity (C); Entropy (D).

and after six months, which indicates that there is an increase of masticatory efficiency at the beginning with slight during the course of observing the edentulous subjects.

The Entropy values are shown on the graph in Fig 4D. Entropy is in correlation with Uniformity and confirms the claims that participants improve their masticatory efficiency after three months and that, by the sixth month, there are only improvements in the fineness of chewing.

The VHO values were lowest for the edentulous participants after three months, which is in correlation with the finding that adequate chewing leads to well-mixed colours and a low VOH [21]. In addition, as seen in Fig 5, the highest VOH results were found in poorly mixed colours through deficient chewing, whereas in this study they were found at the beginning of the period of adaptation.

## Clinical implications

Complete denture wearers are the worst case scenario for chewing gum application. This means that only small changes can be expected—but were detected using DTIA. The applied

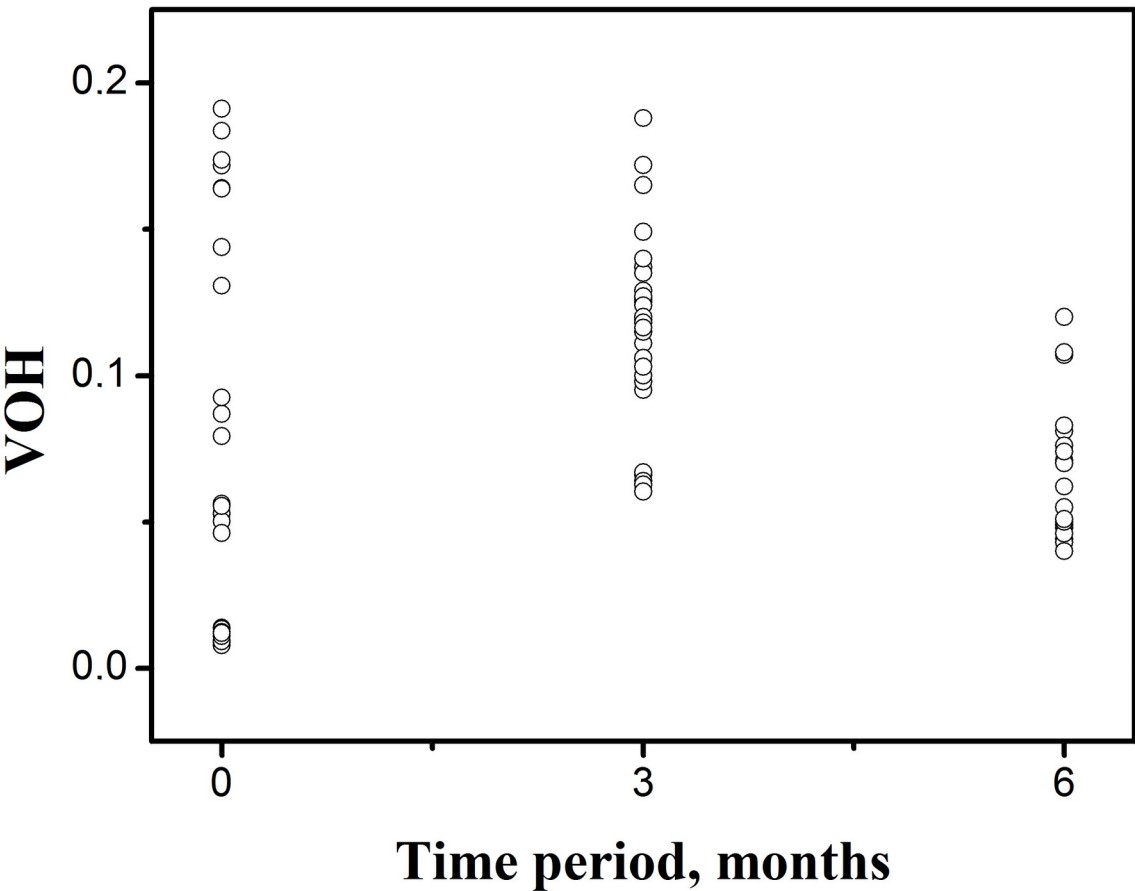

**Fig 5. VOH values of colorimetric analysis for each patient during the observation period (at the baseline, after three months, after six months).**

method enabled observation of differences in the chewed samples within the group, but also the comparison of the obtained samples at defined time intervals. The analysed DTIA features confirmed a gradual improvement of masticatory efficiency in the study group over the evaluation period. Therefore, the improvement of the masticatory function in relation to the mixing ability of two-coloured chewing gum could be traced by monitoring changes in the values of DTIA variables. The method has proven to be reliable in complete denture wearers and should be tested with a different number of chewing cycles, as well as in dentate subjects and / or different dental restoration wearers.

## Conclusion

Accepting the discussed weakness of the study, it may be presumed that DTIA can be considered a reliable tool for assessing changes in the masticatory efficiency through mixing behavior of two-coloured chewing gum samples. A hypothesized increase of masticatory efficiency was observed by DTIA, which thereby seems to be sensitive enough to detect improvement in chewing ability. This is most predictable by monitoring DTIA parameters such as contrast, and homogeneity, but not uniformity. Further studies shall clarify the reproducibility and threshold of values to better interpret the data from DTIA alone, as well as in hand with VHO.

## Supporting information

**S1 Appendix.**
(PDF)

**S1 Dataset.**
(XLSX)

## Author Contributions

**Conceptualization:** Aleksandra Milić Lemić.

**Formal analysis:** Katarina Rajković, Biljana Miličić.

**Methodology:** Aleksandra Milić Lemić, Katarina Radović, Mirjana Perić.

**Project administration:** Mirjana Perić.

**Resources:** Rade Živković.

**Writing – original draft:** Aleksandra Milić Lemić.

**Writing – review & editing:** Aleksandra Milić Lemić.

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
