## [Editor Report · Decision Letter 0]

18 Jun 2020

PONE-D-20-18199

Within diabetic subject evaluation of masticatory efficiency using digital image texture analysis

PLOS ONE

Dear Dr. Milić Lemić,

Thank you for submitting your manuscript to PLOS ONE. After a first assessment I have to ask you for some adaptations before I am able to send it to potential reviewers.

My major points are the following:

a) you state in the introduction that edentulism is a prevalent problem

- please give references. Actually edentulism is decreasing across Europe - but

of course not world wide.

b) your number of patient (N=19) with DM is rather low for sophisticated statistical

methods and p-value fishing. Thus, I invite you to better depict the graphical data

with representation of the deviations by single dots of each measurement to allow a

c) more qualitative approach to your results and findings. This could include to track

the performance (or measurements) of each individual from T0->T2 graphically.

d) I am missing the STROBE checklist as well as data about the drop outs, exclusions

and reasons for exclusion.

e) You assessed the Kapur score - please include the data to your findings and as

highlighted in c) contrast the distributions of this score to the findings in your

variables under evaluation.

f) These measures may enable a better understanding of the results especially the

divergence between VOH and contrast from T0->T2

g) please only enclose high resolution images in line art (all Figures in TIFF are

depicted awfully - please check conversions or extraction paths)

h) please help the reader to understand the variables more clearly.

For me, the differences in variables under observation are quite small. How does

dentate patients perform in this test? This is crucial because statistical significance has

nothing to do with relevance.

i) To understand deviations please give the data of both cycles (I assume but you didnt mentioned that you

used the mean of both chewings) and perform an intra-individual analysis

about the reproducibility at each T. Therefore, I recommend to work rather with differences and deviations from

the common mean than with extraorinary statistical methods.

Finally I am looking forward to reassess and handle your manuscript as far as you are

willing to adjust the points raised by my side,

We look forward to receiving your revised manuscript.

Kind regards,

Fabian Huettig, DMD, Ph.D.

Academic Editor

PLOS ONE

Journal Requirements:

2. Your ethics statement must appear in the Methods section of your manuscript. If your ethics statement is written in any section besides the Methods, please move it to the Methods section and delete it from any other section. Please also ensure that your ethics statement is included in your manuscript, as the ethics section of your online submission will not be published alongside your manuscript.

3. Please upload a new copy of Figure 3 as the detail is not clear. Please follow the link for more information: https://blogs.plos.org/plos/2019/06/looking-good-tips-for-creating-your-plos-figures-graphics/
---

## [Author Response · Author response to Decision Letter 0]

18 Aug 2020

a) you state in the introduction that edentulism is a prevalent problem - please give references. Actually edentulism is decreasing across Europe – but of course not world wide.

As you have pointed the referencse are added in the text, and the paragraph was rewritten. 

b) your number of patient (N=19) with DM is rather low for sophisticated statistical methods and p-value fishing. Thus, I invite you to better depict the graphical data with representation of the deviations by single dots of each measurement to allow a

Our statistician has made changes as you requested. 

Also we were aware of the small sample size at the beginning and addressed it in the Discussion section: “The limitations of this study may be in the small sample size, as well as in the shortcomings of chewing tests. However, evaluation of the literature dealing with similar topics revealed the optimal sample sizes. (34.Van der Bilt A, Speksnijder CM, De Liz Pocztaruk R, Abbink JH. Digital image processing versus visual assessment of chewed two-colour wax in mixing ability tests. Journal of Oral Rehabilitation.2012; 39: 11–17.

35.Miyaura K, Morita M, Matsuka Y, Yamaashita A, Watanabe T. Rehabilitation of biting abilities in patients with different types of dental prostheses. Journal of Oral Rehabilitation. 2000;27: 1073–1076.

36.Gunne H S, Wall A K. The effect of new complete dentures on mastication and dietary intake. Acta Odontologica Scandinavica. 1985;43: 257-268)

c) more qualitative approach to your results and findings. This could include to track the performance (or measurements) of each individual from T0->T2 graphically.

Our statistician has made changes as you requested.

d) I am missing the STROBE checklist as well as data about the drop outs, exclusions and reasons for exclusion. 

Beginning of the Materials and method section was rewritten and modified accordingly in order to present inclusion and exclusion criteria more clearly.

e) You assessed the Kapur score - please include the data to your findings and as highlighted in c) contrast the distributions of this score to the findings in your variables under evaluation.

The paragraph was rewritten since you revealed the lack of clearness. The Kapur scoring was done during the selection of the participants in order to eliminate potential confounders that may influence the results like improper anatomy of the bearing tissues and inadequate clinical quality of the dentures.

f) These measures may enable a better understanding of the results especially the divergence between VOH and contrast from T0->T2

The values of VOH and contrast are not supposed to be correlated but both show similar trend of diminution proposing the fact that there is improvement of masticatory efficiency from T0-T2. Smaller values mean better color mixing and thorough mastication respectively. 

g) please only enclose high resolution images in line art (all Figures in TIFF are depicted awfully - please check conversions or extraction paths)

The Figures were corrected as you requested.

h) please help the reader to understand the variables more clearly. For me, the differences in variables under observation are quite small. How does dentate patients perform in this test? This is crucial because statistical significance has nothing to do with relevance.

In the Methods section we included explanation of the variables in order to have deeper insight into the study results.

i) To understand deviations please give the data of both cycles (I assume but you didnt mentioned that you used the mean of both chewings) and perform an intra-individual analysis about the reproducibility at each T. Therefore, I recommend to work rather with differences and deviations from the common mean than with extraorinary statistical methods.

Part of the manuscript involving statistics analysis was rearranged. As a result of that new Tables 1 & 2 are introduced, and the results are better graphically presented in Figures 4 & 5.

---

## [Decision Letter · Decision Letter 1]

20 Oct 2020

PONE-D-20-18199R1

Within diabetic subject evaluation of masticatory efficiency using digital image texture analysis

PLOS ONE

Dear Dr. Milić Lemić,

Thank you for submitting your manuscript to PLOS ONE. After careful consideration, we feel that it has merit but does not fully meet PLOS ONE’s publication criteria as it currently stands. Therefore, we invite you to submit a revised version of the manuscript that addresses the points raised during the review process.

In order to provide you an atleast timely answer after having a hard time to

find reviewers for your manuscript, I'd like to invite you to adapt your manuscript

towards the points raised by reviewer #1. I am following these points and stay with

him to rather adjust your "story" and to reperform ex ante calculations and to

provide assumptions for your calculations in order to validate and present your method

rather than the data itself (story lost).

If you do not feel comfortable with this recommendation, I suggest that you retract the

paper and hand it to a dental journal for publication.

We look forward to receiving your revised manuscript.

Kind regards,

Fabian Huettig, DMD, Ph.D.

Academic Editor

PLOS ONE

Additional Editor Comments (if provided):

Dear Dr. Lemic,

in order to provide you an atleast timely answer after having a hard time to

find reviewers for your manuscript, I'd like to invite you to adapt your manuscript

towards the points raised by reviewer #1. I am following these points and stay with

him to rather adjust your "story" and to reperform ex ante calculations and to

provide assumptions for your calculations in order to validate and present your method

rather than the data itself (story lost).

If you do not feel comfortable with this recommendation, I suggest that you retract the

paper and hand it to a dental journal for publication.

Reviewers' comments:

Reviewer's Responses to Questions

**Comments to the Author**

1. If the authors have adequately addressed your comments raised in a previous round of review and you feel that this manuscript is now acceptable for publication, you may indicate that here to bypass the “Comments to the Author” section, enter your conflict of interest statement in the “Confidential to Editor” section, and submit your "Accept" recommendation.

Reviewer #1: (No Response)

2. Is the manuscript technically sound, and do the data support the conclusions?

Reviewer #1: Partly

3. Has the statistical analysis been performed appropriately and rigorously? 

Reviewer #1: Yes

4. Have the authors made all data underlying the findings in their manuscript fully available?

Reviewer #1: Yes

5. Is the manuscript presented in an intelligible fashion and written in standard English?

Reviewer #1: Yes

6. Review Comments to the Author

Reviewer #1: 1) The overall intention of this paper remains unclear due to the mix of all the paramenters: diabetic patients, full dentures and a new method to evaluate chewing efficiency.

1a) the used technology was used first time according to the authors - is the method valid and robust? After having measured all the different data - uniformity, contrast, homogeneity, entropy - contrast turned out to be the best marker - but this is post hoc and was not predefined as hypothesis. The authors´conclusion is focused only on that point, diabetes and full dentures are completly ignored in the conclusion. The goal of the study was to see how their exposure to new CDs affects their masticatory performance as treatment outcome (line 72-73). It seems, that this focus was lost somehow.

1b) it remains unclear, why diabetic patients have been selected - is there a difference to non-diabetic patients in chewing performance and/or adaptave capacity to new CDs? So, it is difficult to understand the clinical implications. A specific diabetic related altered learning period in diabetic patients cannot be identified in the conclusion. So, the lake of a control group is critical, especially in combination with the low number of participants.

2) Statistics: there are no information about a repeated measurement design.

3) Age: the range 55 to 80 is wide, and in accordance to the literature (e.g. Peyron MA, Blanc O, Lund JP, et al. Influence of Age on adaptability of human mastication. JNeurophys 2004; 92;773-9 https://doi.org/10.1152/jn.01122.2003), age is an influencing factor. A subgroup analysis for different age groups (at least two) should be included. It is an important influencing factor, that the adaptive capacity is completely different in elderly compared to younger patients groups.

4) Why the chewing test was performed on the preferred chewing side? Why not on both sides? Was preferred chewing side checked again at T1 and T2 - changes can be assumed in adequate and symmetric CD.

Overall:

The parameters diabetic patients, full dentures and a new method are not clearly separated and specified - so the interested clinician might expect an information regarding diabetes or full denture masticatory performance, and not a methodologic study on a new method.

Recommandation:

Clear focus on the new mehtod or on DM in combination CD. Maybe two split into two separated papers?

7. PLOS authors have the option to publish the peer review history of their article (what does this mean?). If published, this will include your full peer review and any attached files.

Reviewer #1: No

---

## [Author Response · Author response to Decision Letter 1]

30 Nov 2020

Response to the Reviewer

Reviewer #1: 1) The overall intention of this paper remains unclear due to the mix of all the paramenters: diabetic patients, full dentures and a new method to evaluate chewing efficiency.

1a) the used technology was used first time according to the authors - is the method valid and robust? After having measured all the different data - uniformity, contrast, homogeneity, entropy - contrast turned out to be the best marker - but this is post hoc and was not predefined as hypothesis. The authors´conclusion is focused only on that point, diabetes and full dentures are completly ignored in the conclusion. The goal of the study was to see how their exposure to new CDs affects their masticatory performance as treatment outcome (line 72-73). It seems, that this focus was lost somehow.

Thank you very much for your comment, beaacuse it forced us to thoroughly go through the text and make substantial corrections. Firstly as you suggested we focused on the novel method and greatly rewrote the introduction with specific ephasize on the existing methods for evaluating masticatory efficiency. Further we added some desctirption regarding DTIA and its contemporary use in dental research so far. As you pointed, we introduced new aims with special focus on evaluating texture features in masticatory efficiency follow up, measuring chewing ability improvement and justifying the use of DTIA in masticatory efficiency evaluation. 

1b) it remains unclear, why diabetic patients have been selected - is there a difference to non-diabetic patients in chewing performance and/or adaptave capacity to new CDs? So, it is difficult to understand the clinical implications. A specific diabetic related altered learning period in diabetic patients cannot be identified in the conclusion. So, the lake of a control group is critical, especially in combination with the low number of participants.

As pointed in the Recommendations: Clear focus on the new mehtod or on DM in combination CD. Maybe two split into two separated papers?

We rewrote our manuscript with completely different focus, abandoning diabetic patients and focusing only on thourough research about the texture analysis and methodical approach with the chewing gum testing. However, we felt that new wearers are representative froup for our methodological approach since they as a new wearers certainly need more time to learn to cope with new dentures. Therefore, we were able to follow them in adequate period of time, providing our method to express itself in full. 

2) Statistics: there are no information about a repeated measurement design.

Every participant chewed two chewing gum samples, according to the described procedure (line 159) in methodology section. And In Table 1 each parameter was presented with test-retest values.

3) Age: the range 55 to 80 is wide, and in accordance to the literature (e.g. Peyron MA, Blanc O, Lund JP, et al. Influence of Age on adaptability of human mastication. JNeurophys 2004; 92;773-9 https://doi.org/10.1152/jn.01122.2003), age is an influencing factor. A subgroup analysis for different age groups (at least two) should be included. It is an important influencing factor, that the adaptive capacity is completely different in elderly compared to younger patients groups.

We are completely aware of the fact you were pointing but age 55 was completely writing mistake, since we at the first place focused on older subjects our inclusion criterion was age over 65, which was written in the methodology section.

4) Why the chewing test was performed on the preferred chewing side? Why not on both sides? Was preferred chewing side checked again at T1 and T2 - changes can be assumed in adequate and symmetric CD.

The used methodology about mixing ability test with chewing double coloured chewing gum was adopted from the literature. 15. Schimmel M, Christou P, Herrmann F, Mȕller F. A two-colour chewing gum test for masticatory efficiency: development of different assessment methods. Journal of Oral Rehabilitation. 2007; 34: 671–678. In order to completely create the real life situation we accepted the proposed the methodology.

Overall:

The parameters diabetic patients, full dentures and a new method are not clearly separated and specified - so the interested clinician might expect an information regarding diabetes or full denture masticatory performance, and not a methodologic study on a new method.

Thank you very much for your comments, because you opened my eyes and I have written the text in completely different manner. Sincerely hope that this new version of the mansucript will satisfy your expectations, because we really put a great effort on it. Of course the merit is on you, but hope we deserved it.

Response to the academic editor

I'd like to invite you to adapt your manuscript towards the points raised by reviewer #1. I am following these points and stay with him to rather adjust your "story" and to reperform ex ante calculations and to

provide assumptions for your calculations in order to validate and present your method rather than the data itself (story lost).

We accepted with great pleasure each point raised by yourself and the reviewer, and are enclosing the new version of the manuscript with changed „ story" and detailed focus on the method implemented. The introduction section is now completely rewritten, with different aims and issues raised. Further text is changed accordingly to meet the new approach in the manuscript. Also we have made changes to the title to better match the text itself. 

Thanks to you we hope that our new version of the manuscript possess merit for the PLOS ONE journal.

---

## [Decision Letter · Decision Letter 2]

22 Jan 2021

PONE-D-20-18199R2

The use of Digital texture image analysis in determining the masticatory efficiency outcome

PLOS ONE

Dear Dr. Milić Lemić,

Thank you for submitting your manuscript to PLOS ONE. After careful consideration, we feel that it has merit but does not fully meet PLOS ONE’s publication criteria as it currently stands. Therefore, we invite you to submit a revised version of the manuscript that addresses the points raised during the review process.

**The manuscript improved, but some major aspects are still lacking to allow a**

**publication. In case of a sufficiently revised resubmission covering all points/comments of the reviewers, I am willing to accept the manuscript in an academically sound structure and in line**

**with the publication criteria (please adhere to raw data submission as well) upon Editorial decision.**

We look forward to receiving your revised manuscript.

Kind regards,

Fabian Huettig, DMD, Ph.D.

Academic Editor

PLOS ONE

Additional Editor Comments (if provided):

The manuscript improved, but some major aspects are still lacking to allow a

publication. In case of a sufficiently revised resubmission covering all points

I am willing to accept the manuscript as academically sound and in line

with the publication criteria (please adhere to raw data submission as well) upon

Editorial decision.

Reviewers' comments:

Reviewer's Responses to Questions

**Comments to the Author**

1. If the authors have adequately addressed your comments raised in a previous round of review and you feel that this manuscript is now acceptable for publication, you may indicate that here to bypass the “Comments to the Author” section, enter your conflict of interest statement in the “Confidential to Editor” section, and submit your "Accept" recommendation.

Reviewer #1: All comments have been addressed

Reviewer #2: (No Response)

2. Is the manuscript technically sound, and do the data support the conclusions?

Reviewer #1: Yes

Reviewer #2: Partly

3. Has the statistical analysis been performed appropriately and rigorously? 

Reviewer #1: Yes

Reviewer #2: Yes

4. Have the authors made all data underlying the findings in their manuscript fully available?

Reviewer #1: Yes

Reviewer #2: No

5. Is the manuscript presented in an intelligible fashion and written in standard English?

Reviewer #1: Yes

Reviewer #2: No

6. Review Comments to the Author

Reviewer #1: Now, the revised manuscript has a clear red line and is therefore adequately understandable.

Recommended adaptations:

a) Table 1: several Spaces missing in column Variables

b) Table 2: Spelling mistake: Entropy

c) Table 1 and Table. 2: the * to highlight significance can be ommitted (to relieve the content of the tables). The p-value is mentioned in the text (line 220 and 221)

d) Table 1 and Table 2: p<0,001 instead of p=0,000 (I gues, that the probability of error is not really zero)

e) Table 1 and Table 2: the meaning of (1) and (2) remains unclear, allthough it seems to be essential for the understanding of the statistical analysis. (1) = DTIA?; (2) = VOH?. Recommendation: no coding, but insertion of the abbreviations.

f) Table 1 and Table 2: the number should be explained in the text - or I havn´t found it.

g) The sentence page 6, lines 127 to 129 should be eliminated or rewritten, otherwise this passage is confusing and falling out of context

The data used to perform the statistical analysis are not explained. So the meaning of those data remains unclear for a reader without a specific background knowledge for DTIA and/or VOH technology. I recommend to add a short paragraph in Material&Method section (or to add a explaining subtext to the tables.

Reviewer #2: Dear authors,

your revision pushed the manuscript into a more readable and understandable form/ structure.

However it is lacking a sufficient scientific English, especially the punctation and construction of sentences should be revised by a native speaker. Basically, there should be one sentence for one thought/statement. Some recommendations were done by me within the word file.

Since your data evaluation follows one possible approach and the raw data are not submitted within the supplement, I exspect that you resubmit a SPSS or CSV or at least XLS file with the data set from the 228 images and all variables.

I suggest a major revision, because the manuscript should be orded as it is the standard. I commented within the word file which passages should be shifted. Finally, the discussion should be revisited and more balanced between your novel findings and the established standard (VHO). This encompasses the reflection of absolute and relative changes despite of the p-value hype.

7. PLOS authors have the option to publish the peer review history of their article (what does this mean?). If published, this will include your full peer review and any attached files.

Reviewer #1: **Yes: **Gregor Slavicek

Reviewer #2: No

---

## [Author Response · Author response to Decision Letter 2]

2 Mar 2021

Dear Reviewers:

Thank you for the opportunity to revise our manuscript, "The use of Digital texture image analysis in determining the masticatory efficiency outcome" (PONE-D-20-18199R2) 

We appreciate the time and effort that the reviewers dedicated to providing valuable feedback on our manuscript. We also appreciate the careful review and constructive suggestions. We have been able to incorporate changes to reflect most of the suggestions made by the reviewers. It is our belief that the manuscript is substantially improved after making the suggested edits.

Following this letter are our point-by-point responses to the reviewers’ concerns in italics, including how the text was modified. The changes made in the manuscript are highlighted in yellow.

Reviewer #1: 

Now the revised manuscript has a clear red line and is therefore adequately understandable.

Thank you very much. Hope that the manuscript is even better after these modifications. 

Recommended adaptations:

a) Table 1: several Spaces missing in column Variables

Changed

b) Table 2: Spelling mistake: Entropy

Changed

c) Table 1 and Table 2: the * to highlight significance can be omitted (to relieve the content of the tables). The p-value is mentioned in the text (line 220 and 221)

Thank you for this, as you recommended we omitted * in the Tables.

d) Table 1 and Table 2: p<0,001 instead of p=0,000 (I guess that the probability of error is not really zero)

Changed

e) Table 1 and Table 2: the meaning of (1) and (2) remains unclear, although it seems to be essential for the understanding of the statistical analysis. (1) = DTIA?; (2) = VOH?. Recommendation: no coding, but insertion of the abbreviations.

Thank you for your suggestion. We have inserted the meaning of (1) and (2) into the Tables and changed them accordingly. Didn`t quite understand what you meant by "no coding, but insertion of the abbreviations"

f) Table 1 and Table 2: the number should be explained in the text - or I haven´t found it.

We mentioned the table number in the text.

g) The sentence on page 6, lines 127 to 129 should be eliminated or rewritten; otherwise this passage is confusing and falling out of context. The data used to perform the statistical analysis are not explained. So, the meaning of those data remains unclear for a reader without specific background knowledge of DTIA and/or VOH technology. I recommend you to add a short paragraph in Material & Method section (or to add an explaining subtext to the tables).

Thank you for your kind notice. The sentence was omitted, and changes were made as you suggested. In the MM section we added the meaning of all DTIA and VOH variables, and our statistician author modified the paragraph explaining the statistical analysis.

Reviewer #2: 

- Dear authors, your revision pushed the manuscript into a more readable and understandable form/ structure. However, it is lacking sufficient scientific English, especially the punctuation and construction of sentences should be revised by a native speaker. Basically, there should be one sentence for one thought/statement. Some recommendations were given by me within the word file. 

Thank you very much for this comment. The manuscript was revised by a professional agency for academic editing. We are adding the proof as a supplementary file.

-Since your data evaluation follows one possible approach and the raw data are not submitted within the supplement, I expect that you resubmit a SPSS or CSV or at least XLS file with the data set from the 228 images and all variables.

During the first submission we submitted the XLS file and SPSS data in the supplementary files as directed by the journal policy. We will do it again in case they are missing.

-I suggest a major revision, because the manuscript should be ordered as it is the standard. I commented within the word file which passages should be shifted. 

Thank you very much for your kindness. We followed the recommendations you inserted within the word file and made changes accordingly.

We also implemented the corrections and suggestions you made within the word file

-Finally, the discussion should be revisited and more balanced between your novel findings and the established standard (VHO). This encompasses the reflection of absolute and relative changes despite of the p-value hype.

As you pointed out, analyzing masticatory efficiency in complete denture wearers is the worst case scenario, and there are not many references using the colorimetric method that investigated ME in CDs wearers. We correlated the studies we analyzed to our findings and broadened the discussion in that way.

-Please restructure your abstract according to the paper and your mentioned aims (i-iii), give the variables first and thereafter the results and conclusion.

Thank you for this suggestion. We have structured the abstract in more proper way. 

- Please describe how this was done – it is important for the reproducibility of your study.

The M&M section was rewritten, thanks to your comment. However, the methodology and procedure for the preparation of gum samples were adopted from the literature and we cited it properly for possible reproducibility.

-Please give the software and operation, as well as if adjustments were set for this conversion.

Thank you for this comment. It was a mistake not providing proper software details. After consulting the author KR who performed the DTIA and VOH, we made some modifications to the text as you suggested. There were no extra manual adjustments of the images, and she relied only on the software to automatically perform the analyses. That way we excluded any manual intrusion in the process. 

-Please add for each variable what a clinical relevant absolute or relative change would be.

Since this was our first DTIA using and following the references regarding the method (there were not many), we decided to follow the changes of the uniformity, contract, entropy and homogeneity. Like other authors, we were not able to give exact values for each of them, but rather to discriminate them during the follow-ups. Of course, implementing the DTIA in dentate subjects might provide more insight into the absolute and relative change. That is our plan for future studies. As far as our edentulous subjects are concerned, for now DTIA has proven to be reliable, providing more detailed analyses than VOH.

-Please describe how you handled the data from both sides of the specimen from both specimens per patient and point in time. This should be described in the subheadings below as well. If you decided for one side and one specimen only – describe why and how.

Thank you for pointing to us this unclearness. Our statistician, author BM reorganized the section and wrote it in a more understandable way.

-According to STROBE, you should give a flowchart on how many patients were screened and how many dropped out/ were closed out to which reasons. This is necessary to understand a potential selection bias.

Thank you for this comment. We provided another flowchart.

-The flowchart should start above your 19 patients.

Yes, we created it accordingly. After a detailed review of our initial XLS data, and SPSS data, we, however, noticed that 18 participants were involved in the study and changed that throughout the text. Our colleague also made changes to Table 1, as requested. 

-Please contrast your overall findings from the DTIA with the published standard VOH. You can detect weaknesses and strength in both approaches. This should be described

Thank you very much for this suggestion and raising that issue. We discussed the strength and weakness of DTIA at the beginning of the discussion. However, the colorimetric analysis is well established in the reference and we didn`t find ourselves competent to discuss it more. We added more sentences concerning the findings of DTIA and VOH. 

The Clinical implications section was rewritten as you suggested. Thank you for pointing out.

-No: you hypothesized it - you can draw the conclusion with the reasoning. This is…

Thank you for your great help in formulating the conclusions. The whole section has been changed and we hope it has quality now and that the conclusions are drawn based on the presented data.

---

## [Editor Report · Decision Letter 3]

9 Mar 2021

PONE-D-20-18199R3

The use of digital texture image analysis in determining the masticatory efficiency outcome

PLOS ONE

Dear Dr. Milić Lemić,

Thank you for submitting your manuscript to PLOS ONE. After careful consideration, we feel that it has merit but does not fully meet PLOS ONE’s publication criteria as it currently stands. Therefore, we invite you to submit a revised version of the manuscript that addresses the points raised during the review process.

We look forward to receiving your revised manuscript.

Kind regards,

Fabian Huettig, DMD, Ph.D.

Academic Editor

PLOS ONE

Journal Requirements:

Additional Editor Comments (if provided):

Dear Dr. Lemic,

thank you for your revision. Your manuscript improved; but could you please fix the shortcomings mentioned

below that I can proceed the decision.

a) Table 1: There are two values in each cell (one of it in italics)

please use the headline oder the Table legend to clarify which values there are.

e.g.: "Mean(SD) and Median (IQR) values of the parameter at the baseline, after three and after six months."

 the last row should be "&Significance" instead of "#Significance" - therefore the last column (right) should be #Significance in headline.

b) Figure 3: Please give age and gender of the participants within the step "assessment" and "exclusion"

and name the failed inclusion criteria of N=7 as well. Please add the time intervals of recall next to the three boxes.

c) S2Appendix: Please adjust the Data headline (ln 2) or provide a legend which is in hand with the wording in your manuscript and add the subjects number, gender, and age.

d) ln 332: please change edentulousness to "edentulism" and crosscheck the wording again.

Best regards, Fabian Huettig

---

## [Author Response · Author response to Decision Letter 3]

23 Mar 2021

Dear Editor Dr Fabian Huettig,

Thank you for the opportunity to revise our manuscript, "The use of Digital texture image analysis in determining the masticatory efficiency outcome" (PONE-D-20-18199R3) 

We appreciate the time and effort that you dedicated to strengthening our manuscript pushing us further to improve it. 

We also appreciate the careful review and constructive suggestions of the reviewers. It is our belief that the manuscript is substantially improved after making the suggested edits.

Following this letter are our point-by-point responses to your pointed shortcomings in italics, including how the text was modified. The changes made in the manuscript are highlighted in yellow.

a) Table 1: There are two values in each cell (one of it in italics) please use the headline oder the Table legend to clarify which values there are.

e.g.: "Mean(SD) and Median (IQR) values of the parameter at the baseline, after three and after six months."

-> the last row should be "&Significance" instead of "#Significance" - therefore the last column (right) should be #Significance in headline.

Authors: Thank you for this suggestion. We used the headline as you suggested and added it below the Table legend.

b) Figure 3: Please give age and gender of the participants within the step "assessment" and "exclusion"

and name the failed inclusion criteria of N=7 as well. Please add the time intervals of recall next to the three boxes.

Authors: We named the failed inclusion criteria for the seven subjects as you requested. Unfortunately, we are not able to give proper data concerning age and gender of the participants within the assessment and exclusion step. This information was gathered at the very beginning and since the main outcome of the study was not related to differences in age and gender, we do not have those papers with us anymore. 

-Please add the time intervals of recall next to the three boxes.

Authors: We are afraid we do not understand exactly what you meant. However, we inserted that recalls were after three months. As it was noted in the text, the study lasted for ten months, and not all subject were analysed at the same time.

c) S2Appendix: Please adjust the Data headline (ln 2) or provide a legend which is in hand with the wording in your manuscript and add the subjects number, gender, and age.

Authors: Thank you for this suggestion, we corrected as you requested. 

d) ln 332: please change edentulousness to "edentulism" and crosscheck the wording again.

Authors: It was changed as you requested. 

Authors: We are aware that we were not able to respond to your request and are sorry about that. The information regarding age and gender of the participants within the assessment and the exclusion step are no longer with us. However, we hope that the lack of data concerning the age and gender in the supplementary files might not significantly affect the quality of the manuscript we reached after your guidance and that it will not affect your decision.

---

## [Editor Report · Decision Letter 4]

19 Apr 2021

The use of digital texture image analysis in determining the masticatory efficiency outcome

PONE-D-20-18199R4

Dear Dr. Milić Lemić,

We’re pleased to inform you that your manuscript has been judged scientifically suitable for publication and will be formally accepted for publication once it meets all outstanding technical requirements.

Kind regards,

Fabian Huettig, DMD, Ph.D.

Academic Editor

PLOS ONE

Additional Editor Comments (optional):

Thank you for your prompt and sufficient adaptations of your manuscript.

I am looking forward to seeing it published within PLOS ONE!
---

## [Editor Report · Acceptance letter]

21 Apr 2021

PONE-D-20-18199R4 

The use of digital texture image analysis in determining the masticatory efficiency outcome 

Dear Dr. Milić Lemić:

I'm pleased to inform you that your manuscript has been deemed suitable for publication in PLOS ONE. Congratulations! Your manuscript is now with our production department. 

Kind regards, 

on behalf of

Dr. Fabian Huettig 

Academic Editor

PLOS ONE